# On the Euclidean action of de Sitter black holes and constrained instantons

Edward K. Morvan[1,2][⋆], Jan Pieter van der Schaar[1,2][†] and Manus R. Visser[3][‡]

**1** Institute of Physics, University of Amsterdam, Science Park 904,
PO Box 94485, 1090 GL Amsterdam, the Netherlands
**2** Delta Institute for Theoretical Physics, Science Park 904,
PO Box 94485, 1090 GL Amsterdam, the Netherlands
**3** Department of Theoretical Physics, University of Geneva,
24 quai Ernest-Ansermet, 1211 Genève 4, Switzerland

⋆ e.k.morvanbenhaim@uva.nl , † j.p.vanderschaar@uva.nl , ‡ manus.visser@unige.ch

## Abstract

We compute the on-shell Euclidean action of Schwarzschild-de Sitter black holes, and take their contributions in the gravitational path integral into account using the formalism of constrained instantons. Although Euclidean de Sitter black hole geometries have conical singularities for generic masses, their on-shell action is finite and is shown to be independent of the Euclidean time periodicity and equal to minus the sum of the black hole and cosmological horizon entropy. We apply this result to compute the probability for a nonrotating, neutral arbitrary mass black hole to nucleate spontaneously in empty de Sitter space, which separates into a constant and a "non-perturbative" contribution, the latter corresponding to the proper saddle-point instanton in the Nariai limit. We also speculate on some further applications of our results, most notably as potential non-perturbative corrections to correlators in the de Sitter vacuum.

 Check for updates

# 1  Introduction

As the cosmological, inverted (and singularity free) version of a black hole geometry, a full understanding of the de Sitter geometry is expected to rely on quantum gravity. Combined with the strong observational evidence in support of an early and late approximate de Sitter phase in our universe, de Sitter remains a fruitful theoretical playground for testing some of the most promising ideas in quantum gravity [1–6]. Of course, the absence of supersymmetry and a holographic dual description implies one has far less control in general. Nevertheless, the recent promising developments in string theory and AdS/CFT addressing black hole unitarity from a bulk perspective [7,8], pointing to a special role for the (low energy effective) Euclidean action, seem concrete and tempting enough to test in a de Sitter environment. As a first step in this direction we will revisit, extend and reinterpret some old results on instantons and Euclidean actions in de Sitter space [9,10].

As is well known, the Euclidean action of de Sitter space reproduces minus the gravitational entropy of the cosmological horizon [11,12]. When considering a nonrotating, neutral black hole in de Sitter space, general (thermodynamical) considerations would suggest that the Euclidean action should separate into two parts, one describing the contribution from the cosmological horizon and the other from the black hole horizon. However, the absence of equilibrium in the Schwarzschild-de Sitter (SdS) case implies that the Euclidean solution is singular, obscuring the meaning of the corresponding on-shell action. So a proper verification and understanding of the standard (thermodynamical) result for the action appears to require a new approach, which might be relevant for an improved understanding of quantum de Sitter in general, and its Hilbert space in particular. We will present what we believe to be a mathematically and internally consistent procedure that produces the expected answer, and provide a suitable physical interpretation.

One surprising aspect of the expected answer for the on-shell action as the sum of two gravitational entropies is that the total entropy decreases as the mass of the black hole increases. The maximum entropy state apparently corresponds to empty de Sitter, whereas the minimum entropy state is achieved by introducing the largest black hole possible, the Nariai limit. This suggests that we can think of black holes in de Sitter space as localized, more organized, constrained, states of the original de Sitter degrees of freedom, explaining the decrease in entropy [13–16]. It is this interpretation of black holes in de Sitter as constrained states that we will implement concretely in the Euclidean action framework. By identifying the Euclidean de Sitter black hole solutions as constrained instantons we can keep track of, and sum over, their contribution to the Euclidean path integral [17].

One of our main results is that we present a completely general derivation showing that the Euclidean action of a general Schwarzschild-de Sitter solution with conical singularities, in arbitrary dimensions $d > 3$, after imposing the (nonlinear) Smarr relation between the black hole and cosmological horizons, is completely independent of the identified temperature, and

equal to minus the sum of the two Gibbons-Hawking horizon entropies [18–21]

$$I_E = -\frac{A_b + A_c}{4G} = -S_{SdS}.$$ (1)

As already mentioned, an important realization in this context is that the general (singular) SdS solution should be interpreted as a constrained instanton, where the mass of the black hole is fixed. The final answer makes intuitive sense and can be related to the standard approach that just involves the regular Nariai instanton [9,10], by noting that the probability to produce a pair of arbitrary mass black holes separates into a constant and non-perturbative part, where the latter is indeed governed by the Nariai limit. In other words, the gravitational path integral that computes the probability to produce an arbitrary mass black hole in de Sitter space picks up a non-perturbative contribution from the Nariai instanton. We show that the pair creation rate per Hubble volume is

$$\Gamma \approx \int_0^{M_N} dM\, e^{S_{SdS}-S_{dS}} \approx \frac{M_N}{S_{dS}-S_N}\left(1 - e^{-(S_{dS}-S_N)}\right),$$ (2)

where $M$ is the mass of the black hole, $M_N$ is the Nariai mass, $S_{dS}$ is the de Sitter entropy, and $S_N$ is the Nariai entropy. This black hole pair creation rate [22–25] is qualitatively similar to recent results by Susskind [26,27], but our method for computing the probability is different since we integrate $e^{S_{SdS}-S_{dS}}$ over $M$, instead of over the difference between the locations of the two horizons $x = (r_c - r_b)/L$. In computing this probability we make use of a property that might seem surprising, namely that the difference between the vacuum de Sitter entropy and the entropy of the Schwarzschild-de Sitter solution, in arbitrary dimensions $d > 3$, is to a very good approximation a linear function of the mass of the black hole.

   In section 2 we derive equation (1) for the on-shell Euclidean action of Schwarzschild-de Sitter spacetime. This is followed up by a short review of constrained instantons and their role in the path integral in section 3, resulting in the expression (2) for the probability to nucleate an arbitrary mass black hole. In the conclusions we summarize our findings and end with some speculations on the implications for de Sitter fragmentation and instanton corrections to correlators in de Sitter space. Throughout we work with units where $\hbar = c = 1$, but we keep Newton's constant explicit, in order to be able to distinguish between perturbative and non-perturbative corrections in $G$.

   *Note added:* near the completion of our paper we became aware of independent related work [28], where it is also argued that Euclidean Schwarzschild-de Sitter is a genuine stationary point of the constrained path integral. Similar ideas were explored a long time ago in [18,19], as well as [20], where in the latter they make use of the Hartle-Hawking wavefunction, but without a proper understanding of the constrained path integral. Only recently did we become aware of very similar results for the on-shell action of Schwarzschild-de Sitter, derived and used however in a different context [21,29]. Our derivation of the on-hell action differs from previous work in that it is covariant, it emphasizes the role of the Smarr relation and it holds for arbitrary dimensions.

## 2   Euclidean action and entropy of Schwarzschild-de Sitter

After reviewing some geometric and thermodynamic properties of Schwarzschild-de Sitter spacetime, we compute the on-shell Euclidean action explicitly. We also analyze the total gravitational entropy of SdS and show that it can be approximated by a linear function of the mass parameter. See [9,10,12,16,26–28,30–48] for a list of previous literature on SdS.

## 2.1   SdS geometry and its non-equilibrium thermodynamics

Schwarzschild-de Sitter or Kottler [49] spacetime is the neutral, static, spherically symmetric solution of the Einstein equation with a positive cosmological constant $\Lambda$. The $d$-dimensional SdS metric in static coordinates is given by

$$ds^2 = -f(r)dt^2 + f^{-1}(r)dr^2 + r^2 d\Omega_{d-2}^2\,, \quad \text{with} \tag{3}$$

$$f(r) = 1 - \frac{r^2}{L^2} - \frac{16\pi GM}{(d-2)\Omega_{d-2}r^{d-3}}\,. \tag{4}$$

Here, $L = \sqrt{(d-1)(d-2)/(2\Lambda)}$ is the de Sitter curvature radius, $G$ is Newton's constant, $M$ is the mass parameter, and $\Omega_{d-2} = 2\pi^{(d-1)/2}/\Gamma[(d-1)/2]$ is the volume of a unit $(d-2)$ sphere. For generality we keep the number of spacetime dimensions $d$ arbitrary in this paper. The SdS metric represents a black hole in asymptotically de Sitter space. Requiring the absence of a naked singularity yields an upper limit on the mass of SdS black holes

$$0 \le M \le M_N\,. \tag{5}$$

The case $M = 0$ corresponds to pure de Sitter space, which has a single (cosmological) event horizon located at $r_0 = L$. For $d \ge 4$ and the values $0 < M < M_N$ the function $f(r)$ has two positive real roots $r_b$ and $r_c$, with $r_b \le r_c$, corresponding to the position of the black hole event horizon and the cosmological event horizon, respectively. As $M$ increases, the black hole horizon increases in size, whereas the cosmological horizon shrinks in size due to the gravitational pull of the black hole. The upper bound $M = M_N$ corresponds to the case where the black hole and cosmological horizon coincide, $r_b = r_c = r_N$, known as the Nariai solution [9,50]. This is the largest possible black hole in de Sitter space. The Nariai mass and horizon radius may be found by solving $f(r_N) = f'(r_N) = 0$, yielding

$$M_N = \frac{d-2}{d-1}\frac{\Omega_{d-2}}{8\pi G}r_N^{d-3}\,, \qquad r_N = L\sqrt{\frac{d-3}{d-1}}\,. \tag{6}$$

We often express SdS quantities in terms of the dimensionless ratio $\mu \equiv M/M_N$, for which the range (5) becomes $0 \le \mu \le 1$. In the Nariai limit the SdS geometry reduces to $dS_2 \times S^{d-2}$, where the curvature radius of two-dimensional de Sitter space is $\hat{L} = L/\sqrt{d-1}$, and the curvature radius of the sphere is equal to $r_N$, which are equal in $d = 4$ [9,10,31,51,52]. In the near-Nariai limit for a spherical reduction of four-dimensional SdS the Einstein action reduces to the action of de Sitter JT gravity, which was studied in [52–54].

Solving $f(r_i) = 0$ in $d$ dimensions provides $d-2$ relations between the $d-1$ roots

$$\sum_{i=1}^{d-1} r_i^n = \begin{cases} 0\,, & \text{for } 0 < n \le d-2 \text{ odd}\,, \\ 2L^n\,, & \text{for } 0 < n \le d-2 \text{ even}\,, \end{cases} \tag{7}$$

which are enough equations to rewrite every root $r_i$ as a function of $r_b$ and/or $r_c$. Further, solving $f(r_b) = f(r_c) = 0$ yields the following expressions for the mass parameter $M$ and the de Sitter radius $L$ in terms of the horizon radii

$$\frac{16\pi GM}{(d-2)\Omega_{d-2}} = \frac{r_c^{d-1}r_b^{d-3} - r_b^{d-1}r_c^{d-3}}{r_c^{d-1} - r_b^{d-1}}\,, \qquad \frac{1}{L^2} = \frac{r_c^{d-3} - r_b^{d-3}}{r_c^{d-1} - r_b^{d-1}}\,. \tag{8}$$

Inserting this into the blackening function gives

$$f(r) = \frac{1}{r_c^{d-1} - r_b^{d-1}}\left[r_c^{d-1}\left(1 - \frac{r^2}{r_c^2} - \frac{r_b^{d-3}}{r^{d-3}}\right) - r_b^{d-1}\left(1 - \frac{r^2}{r_b^2} - \frac{r_c^{d-3}}{r^{d-3}}\right)\right]\,. \tag{9}$$

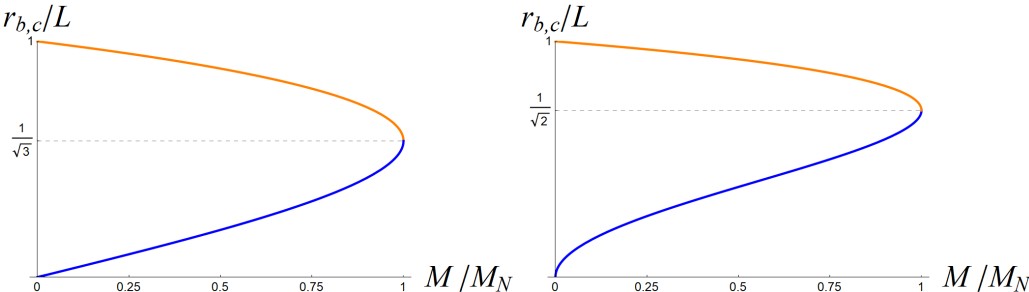

Figure 1: Horizon radii $r_{b,c}$ in units of the de Sitter radius $L$ vs. mass $M$ in units of the Nariai mass $M_N$, where we have set $d = 4$ (left) and $d = 5$ (right). The black hole horizon radius (blue curve) becomes larger as $M$ increases, whereas the cosmological horizon (orange curve) shrinks in size as $M$ increases. For the Nariai solution the horizon radii coincide.

Note in the limit $r_b \to 0$ we recover pure de Sitter spacetime, and the limit $r_c \to \infty$ corresponds to asymptotically flat Schwarzschild spacetime.

In general dimensions there is no closed form expression for $r_b$ and $r_c$ as a function of $M$ and $L$. However, in $d = 4$ one can find the expressions [36–38]

$$r_b = r_N \left( \cos\eta - \sqrt{3}\sin\eta \right), \tag{10}$$

$$r_c = r_N \left( \cos\eta + \sqrt{3}\sin\eta \right), \tag{11}$$

where $\eta \equiv \frac{1}{3}\arccos(M/M_N)$, $r_N = \frac{L}{\sqrt{3}}$, and $M_N = \frac{L}{3\sqrt{3}G}$, and in $d = 5$ we have

$$r_b = r_N \sqrt{1 - \sqrt{1 - M/M_N}}, \tag{12}$$

$$r_c = r_N \sqrt{1 + \sqrt{1 - M/M_N}}, \tag{13}$$

where $r_N = \frac{L}{\sqrt{2}}$ and $M_N = \frac{3\pi L^2}{32G}$. These explicit relations are useful when making plots. In Figure 1 we plot the horizon radii $r_{b,c}$ in four and five dimensions as a function of the mass.

Next we recap the thermodynamics of the SdS black hole solution. Due to thermal radiation emitted from their respective horizons, we can assume that both the black hole and the cosmological horizon have an associated temperature and entropy [12, 55]

$$T_{b,c} = \frac{\kappa_{b,c}}{2\pi}, \qquad S_{b,c} = \frac{A_{b,c}}{4G}. \tag{14}$$

Here, $\kappa_{b,c}$ and $A_{b,c} = \Omega_{d-2} r_{b,c}^{d-2}$ denote the surface gravity and area of the black hole and cosmological horizon, respectively. For arbitrary masses the temperatures $T_b$ and $T_c$ are not the same, so SdS is out of equilibrium in general. Only for the Nariai solution do the temperatures coincide, and are the two horizons in thermal equilibrium. Pure de Sitter space is also in thermal equilibrium (in the Bunch-Davies state), but it has a single (cosmological) event horizon.

The surface gravities are defined with respect to the Killing vector $\xi = \gamma \partial_t$ generating time translations, where $\gamma$ is an arbitrary normalization constant. The standard definition $\xi^\mu \nabla_\mu \xi^\nu = \kappa \xi^\nu$, evaluated at the future black hole and cosmological event horizon, yields $\kappa_{b,c} = \frac{1}{2}\gamma |f'(r)|_{r=r_{b,c}}$. We thus find

$$\kappa_b = \gamma \left( \frac{1}{2}\frac{d-3}{d-2}\frac{16\pi GM}{A(r_b)} - \frac{r_b}{L^2} \right) = \gamma(d-1)\frac{r_N^2 - r_b^2}{2 r_b L^2},$$

$$\kappa_c = \gamma \left( -\frac{1}{2}\frac{d-3}{d-2}\frac{16\pi GM}{A(r_c)} + \frac{r_c}{L^2} \right) = \gamma(d-1)\frac{r_c^2 - r_N^2}{2 r_c L^2}. \tag{15}$$

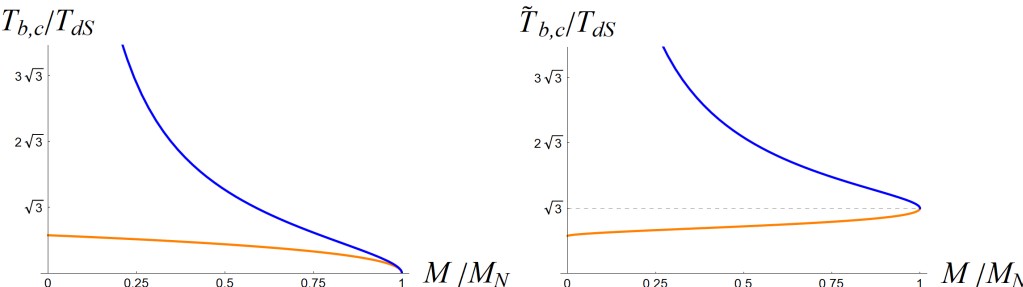

Figure 2: Temperatures $T_{b,c}$ in units of $T_{dS} = 1/2\pi L$ vs. unitless mass parameter $M/M_N$, where we have set $d = 4$. The black hole temperature is shown in blue and the cosmological horizon temperature in orange. The left figure shows the temperatures $T_{b,c} = \frac{\kappa_{b,c}}{2\pi}$, where the surface gravities are defined with respect to the time translation Killing vector $\xi = \partial_t$. The right figure shows the Bousso-Hawking temperatures $\tilde{T}_{b,c} = \frac{\tilde{\kappa}_{b,c}}{2\pi}$, which are normalized with respect to a free-falling static observer, cf. (17). On the left the temperatures go to zero in the Nariai limit, whereas on the right the Nariai temperature is finite $T_N = \frac{\sqrt{3}}{2\pi L}$.

Note that $\kappa_b > \kappa_c$, hence the black hole is hotter than the cosmological horizon. In principle, the normalization factor $\gamma$ can be arbitrary, but two special choices appear often in the literature. The first choice is $\gamma = 1$, for which both surface gravities vanish in the degenerate (Nariai) limit. This normalization seems unphysical since the Nariai black hole is expected to be in thermal equilibrium at a finite temperature, as it is just two-dimensional pure de Sitter space times a sphere. This issue can be remedied by choosing a different value for $\gamma$. For instance, as suggested by Bousso and Hawking, the Killing vector must be normalized on the geodesic orbit of an observer who can stay in place without accelerating (see the Appendix in [10]). This occurs at a fixed radius $r_0$ where the blackening factor attains its maximum, $f'(r_0) = 0$, given by

$$r_0^{d-1} = \frac{d-3}{d-2} \frac{8\pi G M L^2}{\Omega_{d-2}} = \frac{d-3}{2} r_{b,c}^{d-3} \left( L^2 - r_{b,c}^2 \right) = r_N^{d-1} \mu. \tag{16}$$

Plugging this back into the blackening factor gives $f(r_0) = 1 - r_0^2/r_N^2$. Such that surface gravity using the normalization factor $\gamma = 1/\sqrt{f(r_0)}$ becomes

$$\tilde{\kappa}_{b,c} = \frac{\kappa_{b,c}}{\sqrt{f(r_0)}} = \frac{(d-1)r_N^2}{2r_{b,c} L^2} \frac{|1 - r_{b,c}^2/r_N^2|}{\sqrt{1 - r_0^2/r_N^2}}. \tag{17}$$

In the limit $r_{b,c} \to r_N$ we find $\tilde{\kappa}_N = \sqrt{d-1}/L$ (see Appendix B in [52]). Thus considering the free-falling observer normalization gives a non-vanishing surface gravity for the Nariai black hole. In Figure 2 we plot the horizon temperatures for the normalization $\gamma = 1$ (left figure) and $\gamma = 1/\sqrt{f(r_0)}$ (right figure). Finally, we can also expand the normalized surface gravity near the Nariai solution, $r_{b,c} = r_N(1 \mp \tilde{\epsilon}_0)$. To second order in $\tilde{\epsilon}_0$ we find[1]

$$\tilde{\kappa}_{b,c} = \frac{\sqrt{d-1}}{L} \left( 1 \pm \frac{d-2}{3} \tilde{\epsilon}_0 + \frac{23 + d(2d-11)}{18} \tilde{\epsilon}_0^2 \right) + \mathcal{O}(\tilde{\epsilon}_0^3). \tag{18}$$

The minus sign corresponds to the cosmological horizon, whereas the plus sign is associated to the black hole horizon. For $d = 4$ we recover the result by Bousso and Hawking [10], cf. their Eq. (A.16), $\tilde{\kappa}_{b,c} = \sqrt{\Lambda} \left( 1 \pm \frac{2}{3} \tilde{\epsilon}_0 + \frac{11}{18} \tilde{\epsilon}_0^2 \right) + \mathcal{O}(\tilde{\epsilon}_0^3)$ (see also [9]).

---

[1] Here we used that near $r_{b,c} = r_N$ we have $\sqrt{r_N^2 - r_0^2} \approx \sqrt{d-3}|r_N - r_{b,c}| \left[ 1 + \frac{2d-7}{6} \left( \frac{r_{b,c}}{r_N} - 1 \right) + ... \right]$.

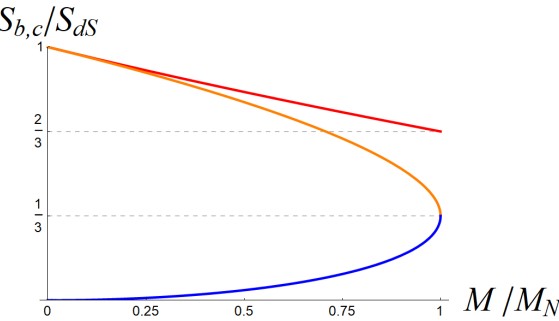

Figure 3: Horizon entropies $S_{b,c}$ in units of the de Sitter entropy $S_{dS}$ vs. unitless mass $M/M_N$, for $d=4$. The blue curve corresponds to the black hole entropy $S_b$, and the orange curve shows the cosmological horizon entropy $S_c$. The sum of the entropies $S_{S_{dS}}$ is approximately a straight line (red curve), and is always less than (or equal to) $S_{dS}$. For the Nariai solution the horizon entropies are both equal to one third of the de Sitter entropy, and their sum is the Nariai entropy $S_N = \frac{2}{3}S_{dS}$. A similar plot appeared in [58].

Both the black hole horizon and the cosmological horizon have an associated first law, which relates the variation of the mass parameter and the horizon area

$$\delta M = T_b \delta S_b, \qquad -\delta M = T_c \delta S_c. \tag{19}$$

Note that the first laws are related, and the associated thermodynamic quantities in them are not independent from each other, *e.g.*, $S_b$ increases as $S_c$ decreases. The minus sign on the right is peculiar and it indicates that the entropy associated to the cosmological horizon goes down as the black hole mass increases. This implies that the entropy of pure de Sitter space $S_{dS}$ is the maximum entropy, and hence dS be interpreted as an equilibrium state with a finite number of degrees of freedom [1].

The question is, however, how the first law for the cosmological horizon can be interpreted as a standard first law of thermodynamics, $dE = T dS$. There are two different interpretations of the minus sign in the right equation in the literature. On the one hand, if we put the minus sign on the right-hand side of the first law, the cosmological horizon has a negative temperature, $T = -T_c$ [56,57]. On the other hand, if we keep the minus sign on the left-hand side, the energy variation becomes negative, $\delta E = -\delta M$ [3]. From this perspective, it seems like increasing the black hole mass parameter decreases the gravitational energy associated to the cosmological horizon, and the sum of the energies is conserved, $E + M = 0$, for small perturbations. The thermodynamic interpretation of the minus sign warrants further investigation, as it is a crucial property of de Sitter space that should also be encoded in a microscopic description of dS.

By adding the variational relations (19) we find the global first law for SdS

$$T_b \delta S_b + T_c \delta S_c = 0. \tag{20}$$

This relation just compares different SdS solutions. If we allow for metric variations away from SdS, by letting the matter stress-energy tensor $T_{\mu\nu}$ be nonzero in the perturbed solution, then the first law for SdS takes the form [12,52]

$$T_b \delta S_b + T_c \delta S_c = -\delta H_\xi^m, \tag{21}$$

where $\delta H_\xi^m = \int_\Sigma \delta T_\mu{}^\nu \xi^\mu u_\nu dV$ is the variation of the matter Killing energy, and $u^\nu$ is the future-pointing unit normal to the spatial section $\Sigma$. The minus sign on the right-hand side

is similar to the one in (19), and it indicates that the horizon entropies $S_{b,c}$ decrease as the matter Killing energy increases.

Finally, let us discuss the generalized Smarr formula for SdS [42]

$$T_b S_b + T_c S_c - \frac{\Theta \Lambda}{(d-2)4\pi G} = 0.$$ 
(22)

Here $\Theta$ is the conjugate quantity to the cosmological constant in an extended version of the first law: $T_b \delta S_b + T_c \delta S_c + \frac{\Theta}{8\pi G} \delta \Lambda = 0$. It can also be defined in a geometric way as the "Killing volume" [52, 57]

$$\Theta = \int_\Sigma |\xi| dV = \gamma \left( \frac{A(r_c)r_c}{d-1} - \frac{A(r_b)r_b}{d-1} \right),$$ 
(23)

where $|\xi| = \sqrt{-\xi \cdot \xi} = \gamma \sqrt{f(r)}$ is the norm of the Killing vector and $dV = \frac{r^{d-2}}{\sqrt{f(r)}} dr d\Omega_{d-2}$ is the proper volume element of the spatial section $\Sigma$ between $r = r_b$ and $r = r_c$. In terms of static coordinates the Smarr formula for SdS reads

$$r_b^{d-2}\kappa_b + r_c^{d-2}\kappa_c - \frac{\gamma}{L^2}(r_c^{d-1} - r_b^{d-1}) = 0,$$ 
(24)

or, equivalently,

$$r_b^{d-2}\kappa_b + r_c^{d-2}\kappa_c - \gamma(r_c^{d-3} - r_b^{d-3}) = 0,$$ 
(25)

where we inserted the equation for $1/L^2$ in (8). It can be verified explicitly using (15) for the surface gravities that this Smarr relation indeed holds for any constant $\gamma$ (which we recall is the normalization of the horizon generating Killing vector $\xi = \gamma \partial_t$).

## 2.2 On-shell Euclidean action

In Euclidean signature the line element of SdS becomes

$$ds^2 = f(r)d\tau^2 + f^{-1}(r)dr^2 + r^2 d\Omega_{d-2}^2,$$ 
(26)

where $\tau = it$ is the Euclidean time. In the Euclidean spacetime there exist conical singularities at the black hole and cosmological horizon. One of the singularities can be removed by requiring the periodicity of the Euclidean time circle to be $\beta = 1/T_{b,c}$, but then the conical singularity at the other horizon remains. Hence there is no completely regular Euclidean section of Schwarzschild-de Sitter spacetime. Only in two special cases does the Euclidean SdS geometry become smooth: the Euclidean de Sitter space is a sphere $S^d$ with radius $L$ and the Euclidean Nariai geometry is $S^2 \times S^{d-2}$, whose curvature radii are $L/\sqrt{d-1}$ and $r_N = L\sqrt{\frac{d-3}{d-1}}$, respectively. Below we consider a generic Euclidean SdS geometry where the periodicity of Euclidean time is equal to an arbitrary inverse temperature, i.e. $\tau \sim \tau + \beta$, such that both conical singularities are present. See Figure 4 for a pictorial representation of Euclidean SdS. Even though Euclidean SdS geometries are not smooth in general, and hence do not represent regular gravitational instantons, we proceed anyway with evaluating the Euclidean action on shell. In fact, it turns out that the on-shell action is finite, since the integral over the conical defects, in low-energy Einstein gravity, is finite. Later in section 3 we give the singular Euclidean SdS spaces a proper interpretation as constrained instantons.

The off-shell Euclidean action of general relativity plus a cosmological constant is

$$I_E = -\frac{1}{16\pi G} \int_{\mathcal{M}} d^d x \sqrt{g}(R - 2\Lambda).$$ 
(27)

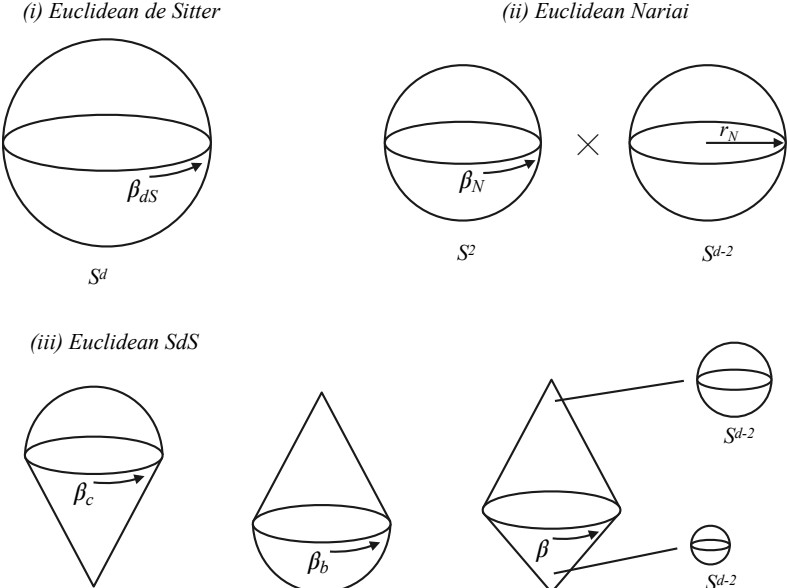

Figure 4: Euclidean SdS geometry. (i) Euclidean de Sitter space is a sphere $S^d$ and the Euclidean time circle has periodicity $\beta_{dS} = 2\pi L$. (ii) Euclidean Nariai space is $S^2 \times S^{d-2}$, where the $S^2$ has a Euclidean time circle with periodicity $\beta_N = 2\pi L/\sqrt{d-1}$ and the $S^{d-2}$ has radius $r_N$. (iii) Euclidean SdS space is a warped product of $S^2$, with one or two conical defects, times $S^{d-2}$. We have drawn three possible Euclidean SdS geometries ("ice cream cones"): on the left we removed the conical singularity at the cosmological horizon by fixing the Euclidean time periodicity to be $\beta_c = 2\pi/\kappa_c$, in the middle we removed the conical singularity at the black hole horizon by fixing the periodicity to be $\beta_b = 2\pi/\kappa_b$, and on the right both conical singularities are present since the periodicity $\beta$ is arbitrary. For these three geometries the radius of the $S^{d-2}$ varies along spatial sections, with the minimal (maximal) sphere corresponding to the black hole (cosmological) horizon.

For Euclidean SdS the Ricci scalar is constant everywhere, except at the conical singularities at $r = r_b$ and $r = r_c$. Hence, it can be written as a sum (see, e.g., [59])

$$R = R_{bulk} + R_{con} = \frac{2d}{d-2}\Lambda + 4\pi\delta(r = r_b) + 4\pi\delta(r = r_c),\tag{28}$$

where $\delta(r = r_{b,c})$ are delta functions at the horizons $r = r_{b,c}$, which correspond to $S^{d-2}$ spheres in the Euclidean geometry. The contribution from the bulk term in the action is

$$I_{E,bulk} = -\frac{\mathcal{V}\Lambda}{(d-2)4\pi G} = -\frac{\beta\Theta\Lambda}{(d-2)4\pi G} = -\frac{\beta\gamma}{8\pi GL^2}(A_c r_c - A_b r_b),\tag{29}$$

where we used $R - 2\Lambda = \frac{4\Lambda}{d-2}$ for SdS in the first equality, and $\Lambda = \frac{(d-1)(d-2)}{2L^2}$ in the last equality. Here, $\mathcal{V} = \int_{\mathcal{M}} d^d x \sqrt{g}$ is the Euclidean spacetime volume, which turns out to be equal to $\mathcal{V} = \beta\Theta$, where $\Theta$ is the Killing volume in Eq. (23). The normalization factor of $\gamma$ appears on the right side of (29) since we assume that $\beta$ is the periodicity of a rescaled Euclidean time $\tau/\gamma$, so that the associated Killing vector in Euclidean spacetimes is $\xi = \gamma\partial_\tau$.

Further, the delta functions in the Ricci scalar are integrable, giving the following contribution to the action

$$I_{E,con} = -\frac{A_b\epsilon_b}{8\pi G} - \frac{A_c\epsilon_c}{8\pi G} = -\frac{A_b}{4G} - \frac{A_c}{4G} + \frac{A_b n_b}{4G} + \frac{A_c n_c}{4G},\tag{30}$$

where $\epsilon_{b,c} = 2\pi(1 - n_{b,c})$ are the deficit angles associated to the conical singularities, *i.e.*, the angle is identified with period $2\pi n_{b,c}$ at the respective horizons. Near the black hole and cosmological horizon the two-dimensional metric takes approximately the Rindler form $ds^2 = \kappa_{b,c}^2 \rho^2 d(\tau/\gamma)^2 + d\rho^2$. Note that the surface gravities $\kappa_{b,c}$ are given by Eq. (15), and they already include a factor of $\gamma$. If the periodicity of the Euclidean time is given by $\tau/\gamma \sim \tau/\gamma + 2\pi/\kappa_{b,c}$, then the conical singularity is removed at the respective horizon, i.e. the polar angle $\phi = \kappa_{b,c}\tau/\gamma$ is identified with $\phi \sim \phi + 2\pi$. This implies that if the periodicity of the Euclidean time is a generic function $\beta$, then the periodicity of the polar angle is $\beta\kappa_{b,c} = 2\pi n_{b,c}$.

Therefore, the total on-shell Euclidean action takes the form

$$I_E(\beta) = I_{E,bulk} + I_{E,con} = \frac{\beta\Omega_{d-1}}{8\pi G}\left[r_b^{d-2}\kappa_b + r_c^{d-2}\kappa_c - \frac{\gamma}{L^2}\left(r_c^{d-1} - r_b^{d-1}\right)\right] - \frac{A_b}{4G} - \frac{A_c}{4G}, \quad (31)$$

or, equivalently, written in a more covariant form,

$$I_E(\beta) = I_{E,bulk} + I_{E,con} = \beta\left[\frac{A_b\kappa_b}{8\pi G} + \frac{A_c\kappa_c}{8\pi G} - \frac{\Theta\Lambda}{(d-2)4\pi G}\right] - \frac{A_b}{4G} - \frac{A_c}{4G}. \quad (32)$$

Importantly, the Euclidean action is the same for any inverse temperature $\beta$. Next, we recognize that the term between square brackets multiplying $\beta$ vanishes due to the Smarr formula (22) for SdS, and thus the Euclidean action becomes minus the sum of the horizon entropies

$$I_{E,SdS} = -\frac{A_b + A_c}{4G}. \quad (33)$$

Note, in particular, that the total action is independent of the normalization $\gamma$ of the Killing vector. Moreover, this resulting Euclidean action holds for any value of the mass $M$. Recently, Draper and Farkas [28] arrived at the same result, by evaluating the Gibbons-Hawking-York boundary terms on infinitesimal boundaries surrounding the two horizons, following the approach in [60].

Previously, the references [9,10] only computed the on-shell action for pure de Sitter space and for the extremal Nariai solution (and for small perturbations away from extremality). Bousso and Hawking [10] found an expression for the total Euclidean action of SdS in $d = 4$, $I_{E,SdS} = -\frac{\mathcal{V}\Lambda}{8\pi G} - \frac{A_b\epsilon_b}{8\pi G} - \frac{A_c\epsilon_c}{8\pi G}$, cf. their Eq. (A.17), however they did not use the Smarr relation for SdS to evaluate the expression further and show that it is equal to minus the total gravitational entropy. At the time they did not consider general Euclidean SdS geometries to be worthwhile studying, as they are not genuine saddle points of the path integral.

For the special case $M = 0$ the Euclidean action reduces to minus the de Sitter entropy

$$I_{E,dS} = -\frac{A(L)}{4G}. \quad (34)$$

For instance in $d = 4$ we find $I_{E,dS} = -\frac{3\pi}{\Lambda G}$, in agreement with the Gibbons-Hawking result [11]. We can also expand the action around pure de Sitter space for small $M$

$$I_{E,ndS} = -\frac{A(L)}{4G} + 2\pi M L + \mathcal{O}(M^2) = -\frac{A(L)}{4G} + (d-2)\frac{A(L)}{4G}\epsilon^2 + \mathcal{O}(\epsilon^4), \quad (35)$$

where we used the fact that the cosmological horizon radius, for a finite positive mass parameter, is smaller than the de Sitter radius, *i.e.*, $r_c = L - \frac{8\pi G M}{(d-2)\Omega_{d-2}L^{d-4}} = L(1 - \epsilon^2)$. The term $2\pi M L$ is the well-known entropy deficit $S_{dS} - S_c$ of the cosmological horizon of SdS to lowest, linear, order in $M$, reproducing the thermal Boltzmann suppression factor [13, 27, 61]. In four dimensions we have $I_{E,ndS} = -\frac{3\pi}{\Lambda G} + \frac{6\pi}{\Lambda G}\epsilon^2 + \mathcal{O}(\epsilon^4)$. In fact, for $d = 4$ we can compute the on-shell Euclidean action *exactly* in $\epsilon$ by solving the relation $r_b^2 + r_c^2 + r_b r_c = L^2$ for

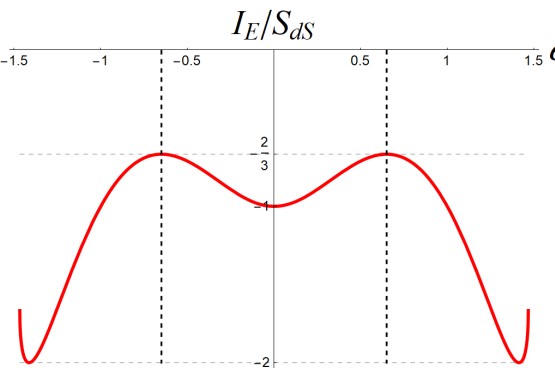

Figure 5: On-shell action $I_{E,SdS}$ of Euclidean SdS in units of $S_{dS}$ vs. the dimensionless parameter $\epsilon = \pm\sqrt{1 - r_c/L}$, where we have set $d = 4$. At $\epsilon = 0$ the action attains a local minimum, which corresponds to Euclidean de Sitter space $S^4$, and at $\epsilon = \pm\sqrt{1 - 1/\sqrt{3}}$ (dashed lines) the action attains a local maximum, corresponding to Euclidean Nariai space $S^2 \times S^2$. At $|\epsilon| = \sqrt{2}$ the action has a global minimum which resembles pure de Sitter space, see Appendix A. There is a negative mode between the de Sitter and Nariai stationary points.

$r_b$ [13, 27]. Inserting the resulting expression for $r_b$ and $r_c = L(1 - \epsilon^2)$ into the Euclidean action $I_{E,SdS} = -\frac{\pi}{G}(r_b^2 + r_c^2) = -\frac{\pi}{G}(L^2 - r_b r_c)$ yields

$$I_{E,SdS} = -\frac{3\pi}{\Lambda G}\left(1 - \frac{1}{2}\left(1 - \epsilon^2\right)\sqrt{1 + 6\epsilon^2 - 3\epsilon^4} + \frac{1}{2}\left(\epsilon^2 - 1\right)^2\right), \quad \text{for} \quad d = 4, \tag{36}$$

where we used $L^2 = 3/\Lambda$. Plotting this expression for the action for fixed values of $L$ and $G$ shows that de Sitter space is a local minimum of the action (at $\epsilon = 0$) and that the Nariai solution is a local maximum (at $\epsilon = \pm\sqrt{1 - r_N/L}$). In Figure 5 we also plotted the action beyond the maxima to see the extremal points more clearly and to connect to the "inside-out" transition by Susskind [26, 27] (see Appendix A for further analysis). In terms of the parameter $\epsilon$ the de Sitter and Nariai clearly correspond to stationary points of the Euclidean action, suggesting that $\epsilon$ can be associated to a specific metric variation of the action.

Furthermore, for the Nariai solution the action becomes equal to minus the Nariai entropy

$$I_{E,N} = -2\frac{A(r_N)}{4G}, \tag{37}$$

which is $-\frac{2\pi}{\Lambda G}$ for $d = 4$, in agreement with [9, 10]. We can of course also decide to expand the on-shell Euclidean action around the extremal Nariai solution, using a different parametrization $r_c = r_N(1 + \tilde{\epsilon})$. If we plug this parametrization into the equation for $1/L^2$ in (8) and solve for $r_b$ to second order in $\tilde{\epsilon}$, then we find $r_b = r_N(1 - \tilde{\epsilon} - \frac{1}{3}(2d - 7)\tilde{\epsilon}^2) + \mathcal{O}(\tilde{\epsilon}^3)$. Near the Nariai solution, in arbitrary dimensions, the action (33) thus becomes

$$I_{E,nN} = -2\frac{A(r_N)}{4G} - \frac{(d-2)^2}{6}2\frac{A(r_N)}{4G}\tilde{\epsilon}^2 + \mathcal{O}(\tilde{\epsilon}^2). \tag{38}$$

In four dimensions we can once again obtain the exact expression

$$I_{E,SdS} = -\frac{3\pi}{\Lambda G}\left(1 + \frac{1}{6}(\tilde{\epsilon} + 1)\left(1 + \tilde{\epsilon} - \sqrt{3(1 - \tilde{\epsilon})(\tilde{\epsilon} + 3)}\right)\right), \quad \text{for} \quad d = 4. \tag{39}$$

Expanding to leading order in the small parameter $\tilde{\epsilon}$ we find $I_{E,nN} = -\frac{2\pi}{\Lambda G} - \frac{4\pi}{3\Lambda G}\tilde{\epsilon}^2 + \mathcal{O}(\tilde{\epsilon}^3)$. This confirms the result by Ginsparg and Perry [9] that the Euclidean action of SdS possesses

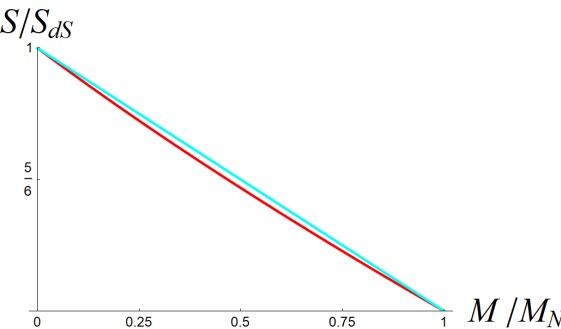

Figure 6: Total gravitational entropy $S_{SdS} = S_b + S_c$ in units of the de Sitter entropy $S_{dS}$ vs. the dimensionless mass $M/M_N$, where we have set $d = 4$. The red curve corresponds to the actual sum of the horizon entropies, whereas the cyan curve is a linear fit. The total entropy is approximately a straight line, which also holds in higher dimensions $d > 4$.

a negative mode in the direction of decreasing black hole mass. However, the particular coefficient of the negative mode in the Euclidean action that we computed differs from Eq. (3.11) in [9] and Eq. (A.18) in [10] (see Appendix B for a more detailed comparison). To conclude, we find that as $M$ is lowered from $M = M_N$ to $M = 0$ the action decreases monotonically from $-2A(r_N)/4G$ to $-A(L)/4G$, implying the existence of a negative mode at the Nariai stationary point. The endpoints $M = 0$ and $M = M_N$ are true extrema of the Euclidean action: de Sitter space is a local minimum and the Nariai solution is a local maximum.

## 2.3 Total gravitational entropy

From the standard definition of the thermodynamic entropy $S = \beta \frac{\partial I_E}{\partial \beta} - I_E$, we find that the total gravitational entropy of SdS is

$$S_{SdS} = \frac{A_b + A_c}{4G}, \tag{40}$$

since the action $I_E$ (33) does not depend on the arbitrary inverse temperature $\beta$. In order to properly interpret this as a thermodynamic entropy, and $\exp(-I_E)$ as the saddle point approximation to a thermal partition function, one should introduce a timelike boundary at some fixed radius $r = R$ [62]. At this boundary either the temperature or the (quasi-local) energy should be fixed depending on whether the thermodynamic ensemble is the canonical or microcanonical ensemble, see [41,48,52,63,64]. In the present paper, however, we are interested in using the Euclidean action to compute the pair creation rate of black holes, and not in defining different thermodynamic ensembles for SdS. We do want to comment, though, that $I_E = -S_{SdS}$ appears to hold in the microcanonical ensemble, as it is valid for abitrary $\beta$, consistent with the interpretation in [28,48].

Next, we analyze the total SdS entropy as a function of the mass parameter $M$, for fixed values of $G$ and $L$ (see Figure 6). Susskind [26,27] has studied the entropy deficit of SdS compared to dS in terms of the difference between the horizon radii $x = (r_c - r_b)/L$. In four dimensions the entropy deficit can be written as $\Delta S = S_{SdS} - S_{dS} = -\frac{1}{3}S_{dS}(1 - x^2)$, but this expression does not generalize in a straightforward way to higher dimensions. Especially when computing the nucleation rate of arbitrary mass black holes the parameter $M$ seems more appropriate to integrate the probability $e^{\Delta S}$ over than the parameter $x$, as we will see in section 3.2.

Before we go into the dependence of $S_{SdS}$ on the mass, let us first discuss why the total gravitational entropy of SdS is equal to the sum of the black hole and cosmological horizon

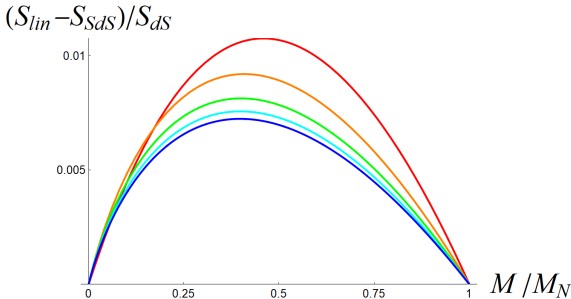

Figure 7: The difference between the linear fit $S_{lin}$ and the actual total entropy $S_{SdS}$, in units of $S_{dS}$ vs. the mass $M$ in units of $M_N$. The different plots correspond to $d = 4$ (red), $d = 5$ (orange), $d = 6$ (green), $d = 7$ (cyan), $d = 8$ (blue).

entropy (see, *e.g.*, [44, 65]). Could we have anticipated the total entropy in Eq. (40)? For instance, why is the total entropy not $S_c$? Since SdS has two Killing horizons, it is natural to expect that its entropy will be the sum of the two horizon areas. Although it is true that the black hole is contained inside the cosmological horizon, this just means there is a (localized) contribution of $S_b$ (related to the number of black hole microstates) to the total entropy of SdS. The other contribution $S_c$ can then be thought of as related to the number of inaccessible and delocalized de Sitter states. In other words, by creating a black hole one has effectively constrained some dS degrees of freedom to a more "organized" localized state [16]. This can perhaps be realized in some matrix quantum mechanics model [13–15, 27]. If this localized, constrained state with energy $E$ does not add any entropy itself, then $S_c$ equals the total entropy, but for a black hole state one should add $S_b$.[2]

Another argument for identifying the sum of the two horizon entropies with the total SdS entropy follows from bounds on the areas of the black hole and cosmological horizon [66–68]

$$A(L) \geq A(r_b) + A(r_c) \geq 2A(r_N). \tag{41}$$

This can also be seen directly from Figure 3. Since the upper bound $A(L) = \Omega_{d-2}L^{d-2}$ and the lower bound $2A(r_N) = 2\Omega_{d-2}r_N^{d-2}$ are related to the entropies of de Sitter space and the Nariai solution, via the standard Bekenstein-Hawking formula,

$$S_{dS} = \frac{A(L)}{4G}, \qquad S_N = 2\frac{A(r_N)}{4G} = 2\left(\frac{d-3}{d-1}\right)^{\frac{d-2}{2}} S_{dS}, \tag{42}$$

it is natural to expect that the sum $A(r_b) + A(r_c)$ also represents the entropy of arbitrary mass SdS solutions. This is indeed what we find in Eq. (40) from an on-shell Euclidean action computation. In $d = 4$ we have $S_{dS} = \frac{\pi L^2}{G}$ and $S_N = \frac{2}{3}S_{dS} = \frac{2\pi L^2}{3G}$. Notably, in the large $d$ limit the Nariai entropy can be expressed as $S_N(d \to \infty) = (2/e)S_{dS}$, providing an upper bound on the relative entropy decrease due to the presence of a maximal size Nariai black hole.

In Figure 6 we plot the total gravitational entropy $S_{SdS}$ as a function of $M$ (red curve). The entropy of the endpoints of the curve at $M = 0$ and $M = M_N$ are, respectively, given by $S_{dS}$ and $S_N$. We observe that $S_{SdS}$ as a function of $M$ appears to be well approximated by a straight line with a negative slope from $M = 0$ to $M = M_N$. We emphasize that we do not claim to fully understand this observation at this point, except that it is likely related to a large $d$ expansion, as we will point out briefly. For now proceeding, we can thus approximate the total entropy

---

[2]As already mentioned earlier, even when the local excitations do not correspond to black holes, for small $E/M_{pl}$, to leading order the entropy difference reproduces the expected Boltzmann suppression $\exp(S_c - S_{dS}) = \exp(-\beta_{dS}E)$ for s-wave Hawking modes of energy $E$.

by the following linear curve

$$S_{lin} = S_{dS} - \frac{S_{dS} - S_N}{M_N} M \,, \tag{43}$$

which corresponds to the cyan curve in Figure 6. This is a remarkably simple and quite accurate approximation to $S_{SdS}$. That is, the difference $\delta$ between the linear fit and the true total entropy

$$\delta \equiv S_{lin} - S_{SdS} \,, \tag{44}$$

is very small indeed. We plot $\delta$ as a function of $M$ in Figure 7, for various number of dimensions, and find that with respect to $S_{dS}$ the difference is always less than 1.1 percent. Moreover, the linear approximation becomes better as $d$ increases. In fact, using Mathematica it can be shown that the difference (and the integrated difference) approaches zero in the limit of an infinite number of dimensions, suggesting that the linear approximation can be explained in terms of a large $d$ expansion. This linear approximation for the total gravitational entropy will turn out to be particularly useful for computing the nucleation rate of arbitrary mass black holes in de Sitter space.

## 3 Euclidean SdS black holes as constrained instantons

As we have seen the Euclidean SdS action, in arbitrary dimensions, can be computed unambiguously, in the presence of the conical singularities, always reproducing the sum of the black hole and cosmological horizon entropy. The necessary existence of at least one conical singularity signals the fact that the SdS state is not in equilibrium, preventing an understanding of the total Euclidean action of SdS as the free energy (in the canonical ensemble). The presence of a conical singularity would naively also appear to obstruct a standard instanton interpretation. However, as anticipated already in [18–20], but worked out in detail in [28], and further corroborated by our results, Euclidean SdS appears to make sense as a semi-classical stationary contribution to the action as long as we introduce a constraint that fixes the mass of the black hole. This allows for a consistent interpretation of the Euclidean SdS solutions in terms of constrained instantons. In the sections below we will elaborate on, and study the consequences of, this interpretation.

### 3.1 Constrained instantons

Standard instanton solutions correspond to a (smooth) stationary point of the Euclidean action, with a negative mode, allowing for a semi-classical saddle-point approximation to the path integral that allows for the understanding of non-perturbative decay modes. In contrast, constrained instantons are stationary solutions of the action only when a particular constraint is imposed. As a consequence their contribution to the path-integral needs to be re-evaluated, which is what we briefly want to review here, mainly based on the useful and elegant summaries in [17, 69].

For the specific case of Euclidean SdS the constraint that needs to be imposed to ensure that the solution is stationary is that the mass should be fixed. That means, as a corollary, that the on-shell action varies under small variations of the metric that effectively change the mass parameter, except in the extreme limits of empty de Sitter and the maximum mass Nariai solution, which are true non-singular stationary points of the Euclidean action of pure Einstein gravity with a positive cosmological constant. Therefore, as also emphasized in [28], the Euclidean SdS solution should be interpreted as a constrained instanton. Ignoring the latter and using the standard saddle point approximation, and subtracting the Euclidean action of

the empty de Sitter background, one would conclude that the probability to spontaneously nucleate a black hole of mass $M$ in de Sitter space is proportional to

$$\Gamma(M) \propto e^{-\left(I_{E,SdS}(M)-I_{E,dS}\right)} = e^{\left(S_{SdS}(M)-S_{dS}\right)} = e^{\Delta S(M)}. \tag{45}$$

This agrees with the intuitive answer expected on thermodynamical grounds in terms of the entropy deficit $\Delta S(M) = S_{SdS}(M) - S_{dS}$, which in the limit of small $M \ll M_{pl}$, where these solutions can strictly speaking no longer be associated to black holes, reproduces the Boltzmann factor [27]. However, as stressed, since the SdS geometry in general contains singularities one cannot automatically assume it corresponds to a stationary point of the action, so we better accurately verify this result, which will involve the introduction of a constraint.

To properly study and understand the contribution of constrained instanton solutions to the Euclidean gravitational partition function, one introduces the constraint in terms of a delta function and integrates over it to rewrite the path integral as follows

$$\int [dg] e^{-I_E[g]} = \int d\mu \int [dg] \delta(\mathcal{C}[g]-\mu) e^{-I_E[g]}. \tag{46}$$

Here $g$ refers to the metric degrees of freedom and although the above expression is completely general, what we have in mind is that the constraint on the metric components is such that it fixes the mass to be $M$, *i.e.*, we will identify $\mu$ with the dimensionless ratio $M/M_N$ and only consider spherically symmetric degrees of freedom. For a specific implementation of these constraints on the metric for SdS we refer to [28], where the trace $k = \frac{d-2}{R}\sqrt{f(R)}$ of the extrinsic curvature on a surface defined by $r = R$ as embedded in a spatial section of SdS is kept fixed. This procedure removes both conical singularities at the cost of introducing a physical (mirror) shell at $r = R$, but leaves the on-shell Euclidean action invariant, and ensures that the solution is stationary in the presence of the constraint. For our purposes here we will not need to introduce a specific constraint functional $\mathcal{C}[g]$. Indeed, no unique choice exists, different choices should all lead to the same result, corresponding to different ways to slice the path integral. In fact, the constraint can also be understood as a consequence of fixing a particular gauge, implying that certain metric components are held fixed, resulting in solutions that are not necessarily stationary with respect to all variations of the metric [69]. The details of a particular constraint will not be important for the general result we are interested in. In the end the different ways to find constrained solutions should be viewed as different ways to 'decompose' the partition function and identify the non-perturbative instanton contributions, in this case associated to the SdS Euclidean geometry. To identify and approximate these non-perturbative contributions we will need the integral representation of the delta function to impose the constraint by means of a Lagrange multiplier term added to the action

$$\int [dg] e^{-I_E[g]} = \int d\mu \int d\lambda \int [dg] e^{-I_E[g]+\lambda(\mathcal{C}[g]-\mu)}, \tag{47}$$

where it should be understood that the integral over the Lagrange multiplier $\lambda$ is parallel to the imaginary axis. This formal rewriting of the general partition function now allows to identify the contributions from constrained instantons in terms of an actual saddle point approximation. For any fixed $\mu$, only allowing $\lambda$ and the metric degrees of freedom $g$ to vary, the stationary points of the new action, including the Lagrange multiplier term, correspond to solutions of the following equations

$$\delta I_E[g] + \lambda \, \delta\mathcal{C}[g] = 0, \quad \mathcal{C}[g] = \mu. \tag{48}$$

The important observation is that the Euclidean SdS geometry should be a solution to this set of equations for non-vanishing $\lambda$, where $\lambda$ equals the (real) eigenvalue of the (first order)

change in the on-shell action under a small variation of the constraint. For any fixed value of $\mu$, we have now identified a true stationary solution and we can use the saddle point approximation, valid at weak gravitational coupling, to estimate the path integral. Note that even though the stationary solution for a particular $\mu$ identifies a real $\lambda$, as long as the integral over imaginary $\lambda$ passes through the real axis at this particular value for $\lambda$ we can use the saddle point approximation to evaluate the integral. So to summarize, by using the constrained instanton framework the presence of conical singularities in the Euclidean SdS geometry do not obstruct a low-energy semi-classical interpretation in terms of a saddle point, as also suggested in [28] and previously anticipated in [18–20], but the appropriate evaluation of the non-perturbative contribution to the gravitational partition function involves an integral over the constraint parameter.

Integrating over the constraint parameter $\mu \equiv M/M_N$ from 0 to 1, one ends up with a semi-classical approximation of the partition function of Einstein gravity with a positive cosmological constant, in the s-wave sector, that should be valid in the regime of weak gravitational coupling. Introducing the function $F(\mu)$ to account for the zero-mode volume and the Gaussian path integral over the second-order fluctuations around the saddle point (the 1-loop determinant), one then arrives at the final result

$$\int [dg] e^{-I_E[g]} \approx \int d\mu\, e^{-I_{E,SdS}} F(\mu). \tag{49}$$

The exponential indeed confirms the expected behaviour, as already alluded to, at fixed mass. It would certainly be of interest to explicitly construct, for a specific constraint functional, the full stationary solution, including the Lagrange multiplier, and then compute $F(\mu)$, generalizing previous work on the de Sitter and Nariai stationary points [70] to arbitrary black hole mass. For now however, we will only rely on the behaviour of the exponential and proceed under the assumption that the function $F(\mu)$ is constant, only affecting the normalization.[3] We will use this saddle point approximation of the gravitational partition function to derive an expression for the probability to spontaneously nucleate a black hole of arbitrary mass in de Sitter space.

## 3.2 Pair creation rate of black holes in de Sitter space

Unlike hot flat space, which exhibits a classical Jeans instability [71], de Sitter space is classicaly stable at the perturbative level [9]. However, semi-classically de Sitter space allows a pair of black holes to nucleate spontaneously. Ginsparg and Perry [9] computed the probability rate of nucleating a pair of Nariai black holes, which is given by $\exp(S_N - S_{dS}) = \exp(-\pi/\Lambda G)$ in $d = 4$ using the standard approach that subtracts the Euclidean action of the background de Sitter spacetime. Since Euclidean Nariai $S^2 \times S^{d-2}$ is the only regular instanton, they did not consider the contribution from arbitrary mass black holes. However, with the constrained path integral method outlined above, it should now be straightforward to compute the probability rate of pair creating an arbitrary mass black hole in de Sitter space.

The semi-classical approximation to the Euclidean path integral (49) is an integral over constrained instanton SdS solutions with mass $\mu$ from $\mu = 0$ to $\mu = 1$. When interested in the black hole nucleation rate with respect to the de Sitter background, we subtract the action of the Euclidean empty de Sitter solution. This can presumably also be interpreted as comparing the probability $\exp(-I_{E,SdS}(\mu))$ for creating a de Sitter black hole with mass $\mu$ from nothing, to the probability $\exp(-I_{E,dS})$ to create empty de Sitter space from nothing, in a Hartle-Hawking wavefunction of the universe approach [10,20]. The upshot is that we can compute the total

---

[3]We note that our qualitative conclusions will be unaffected as long as $F(\mu)$ is a (positive) power law.

probability rate (per Hubble volume) of pair creating any, arbitrary mass, black hole in de Sitter space, starting from (49), as follows

$$\Gamma \approx M_N \int_0^1 d\mu \, e^{-(I_{E,SdS}(\mu)-I_{E,dS})} = M_N \int_0^1 d\mu \, e^{\Delta S}, \tag{50}$$

where $\Delta S = S_{SdS} - S_{dS}$ is the entropy deficit and $\mu = M/M_N$ is the dimensionless mass parameter. To obtain the dimensions of a rate we multiplied the integral with $M_N$, or equivalently, we integrate over $M$ instead of $\mu$. As mentioned already, we have assumed that the zero-mode volume and 1-loop determinant just provide a normalization factor. The integral over $\exp(\Delta S(\mu))$ from $\mu = 0$ to $\mu = 1$ in arbitrary dimensions is hard to perform exactly, but it can be well-approximated using the linear fit[4] for $S_{SdS}$ (43), implying that

$$\Delta S \approx -(S_{dS} - S_N)\mu. \tag{51}$$

Using this linear fit and integrating over the full physical range of $\mu \in \{0, 1\}$ yields, in arbitrary dimensions,

$$\Gamma \approx \frac{M_N}{S_{dS} - S_N}\left(1 - e^{-(S_{dS}-S_N)}\right). \tag{52}$$

One observes that the resulting pair creation rate has a constant contribution (proportional to $G^0$) $M_N/(S_{dS} - S_N)$ and a non-perturbative contribution proportional to $e^{-(S_{dS}-S_N)}$ coming from the Nariai instanton. We emphasize that the constrained instanton contributions are no longer explicitly present in the final result; we integrated over them and in the final answer Eq. (52) only the stationary Nariai instanton appears explicitly, as a consequence of the boundary of the integral. This is consistent with the fact that constrained instantons are not genuine saddle points of the path integral. In the absence of gravity, i.e. in the limit $G \to 0$ or equivalently $(S_{dS} - S_N) \to \infty$, the nonperturbative term vanishes, consistent with expectations. The coefficient in front equals the slope of the linear fit and sets the natural time scale for the decay rate, which is proportional to the de Sitter temperature

$$\frac{M_N}{S_{dS} - S_N} = \frac{1}{2\pi L}\left[\frac{\sqrt{(d-1)(d-3)}}{d-2}\left(\left(\frac{d-1}{d-3}\right)^{(d-2)/2} - 2\right)\right]^{-1}. \tag{53}$$

This should be expected, since the entropy deficit for small $M$ reproduces the thermal Boltzmann factor, which indeed should roughly give an order one probability per unit Hubble time to see a Hawking (s-wave) mode at an energy scale of the de Sitter temperature. The ($d$ dependent) linear fit explains why the coefficient is not exactly equal to the de Sitter temperature $1/2\pi L$. The transition to black hole production, introducing the Nariai high-energy cut-off, further affects the total probability slightly as compared to the result in the absence of gravity ($G \to 0$).

As pointed out the linear approximation gets more accurate as the number of dimensions increases. Indeed, the probability rate in the limit of a large number of dimensions $d \to \infty$ equals

$$\Gamma(d \to \infty) = \frac{M_N}{(e-2)S_{dS}/e}\left(1 - e^{-(e-2)S_{dS}/e}\right) = \frac{M_N}{(e-2)S_N/2}\left(1 - e^{-(e-2)S_N/2}\right). \tag{54}$$

Concentrating on a more explicit expression in four dimensions the pair creation rate is found to be

$$\Gamma(d=4) \approx \frac{M_N(1 - e^{-S_{dS}/3})}{S_{dS}/3} = \frac{M_N(1 - e^{-S_N/2})}{S_N/2} = \frac{\sqrt{\Lambda}}{3\pi}\left(1 - e^{-\frac{\pi}{\Lambda G}}\right). \tag{55}$$

---

[4]As mentioned in section 2.3 the linear fit can likely be understood in terms of a large $d$ expansion and for our purposes here we have ignored corrections at higher order.

Of course, as alluded to before, this probability only applies to the pair creation of black holes when $M > M_{pl}$. In the current universe, assuming the accelerated expansion is caused by a true cosmological constant, the corresponding Boltzmann suppression is enormous. Our result for the pair creation rate is qualitatively similar, but not equal to Susskind's result, cf. Eq. (5.54) in [26] and Eq. (4.27) in [27], which was motivated in a completely different way, starting from the expected thermodynamical entropy difference formula for the pair creation rate of a black hole of a specific mass $M$, and then integrating over the variable $x = (r_c - r_b)/L$, instead of the constraint parameter $\mu$. Comparing there appears to be a sign difference in front of the non-perturbative contribution, but in particular we do not find an additional (perturbative) entropy suppression factor in front of the non-perturbative term. Perhaps the differences can be attributed to the function $F(\mu)$, which we assumed to be independent of $\mu$, and in principle it should be possible to compute this in the constrained instanton formalism. Nevertheless we believe this result is important as a "proof of concept", in that the total probability can be computed from first principles using the constrained instanton formalism, and we do expect the qualitative features to be unchanged in a more complete calculation.

The result, of course, also changes if we do not integrate $\mu$ from 0 to 1, but instead would have used the Planck mass as a starting point. To interpret the integral as a probability rate to find a classical black hole configuration strictly speaking only make sense for masses $M > M_{pl}$, and therefore one should not take into account contributions from smaller masses. However, as long as one properly interprets the equation as a probability rate to observe an arbitrary s-wave configuration in de Sitter, which for masses below the Planck scale should be associated to massive shells instead of black holes, the result should be correct. Concentrating on black holes only, excluding massive shell contributions below the Planck scale from the integral will change the result ever so slightly and introduce another exponential suppression term, replacing the unit factor in between brackets.[5]

## 4 Conclusion

One important conclusion from our work, as also emphasized in [28, 48], is that the on-shell action of the SdS solution can be given a proper interpretation, either as the microcanonical partition function, directly giving the total entropy, or as a constrained instanton that can be used to compute the transition rate to spontaneously nucleate black holes of arbitrary mass in de Sitter space [18–20]. In our case we explicitly computed the on-shell SdS action in the presence of conical singularities, in arbitrary dimensions, by just imposing the Smarr relation that relates the two conical deficits, yielding the expected answer as the sum of the horizon entropies. Once again this provides evidence in support of the idea that black holes in de Sitter should be thought of as constrained (localized) configurations of the available de Sitter degrees of freedom, lowering the entropy [13–15, 27]. An important remaining question in this regard is exactly why, and under what conditions, the presence of conical singularities nevertheless allows for a low-energy interpretation in terms of non-perturbative constrained instanton contributions.

We also made the observation that the entropy deficit, obtained by comparing the on-shell actions of SdS and empty de Sitter, which is the relevant quantity describing the spontaneous nucleation of a black hole, is well approximated by a linear function of the mass $M$. Moreover, the approximation becomes exact in the limit of an infinite number of dimensions. This property was then used to approximate the integral that computes the nucleation rate of a pair of black holes of any mass. Our result generalizes a previous computation [9, 10] of the probability of nucleating a Nariai black hole to arbitrary SdS black holes [18–20] in general

---

[5]We thank Ben Freivogel for a discussion on this point.

dimensions. As also pointed out in [26, 27] using a different approach, the result splits up in a constant ($G^0$) and non-perturbative contribution, where the non-perturbative contribution is governed by the Nariai instanton. From a path integral perspective this is expected, since the Euclidean Nariai limit corresponds to the unique regular, stationary, instanton (apart from the vacuum de Sitter instanton) that should govern the non-perturbative behaviour. We view our result for the probability as a proof of concept, derived under the assumption that there is no additional $\mu$ dependence in the integral coming from the zero-mode volume and 1-loop determinant, and leave a more complete calculation to future work.

A relatively straightforward generalization of interest is the Euclidean action of charged black holes in de Sitter space. Based on some preliminary results we anticipate that the situation will be very analogous, with the Euclidean action given by the sum of the entropies and a similar interpretation in terms of constrained instantons. It would be interesting to compute the probability of the pair production of charged black holes in de Sitter space, and to analyze which proper instantons contribute most [72]. A better understanding of the charged case, in arbitrary dimensions, might also have interesting applications in the context of the Weak Gravity Conjecture in de Sitter [73]. We plan to report on the charged de Sitter black hole results in the near future.

Based on our results we would like to suggest that the interpretation of Euclidean SdS as a constrained instanton, with a nucleation probability given by the entropy deficit, could also be of relevance for the study of de Sitter fragmentation [74,75]. If a (constrained) instanton exists that describes the nucleation of two or more black hole copies, extending the spacelike sections of the Lorentzian geometry, this would correspond to a more direct description of de Sitter fragmentation as compared to the original proposal [74]. There one first nucleates a Nariai black hole, then introduces perturbations to the Nariai geometry that expand, freeze, and collapse to multiple black holes that consequently Hawking radiate and fragment the original de Sitter space. Perhaps a more direct approach in terms of constrained SdS instantons can shed some new light on the total probability for de Sitter fragmentation to occur. An obvious first guess for this probability, suggested in [20], would be to multiply the SdS Euclidean action with a factor $n$ representing the number of copies, i.e.

$$\Gamma[SdS_n] \propto e^{n\,\Delta S}. \tag{56}$$

This can either be understood in terms of the original single SdS probability, simply repeated $n$ times as more Hubble regions appear due to the ongoing exponential expansion, giving $\Gamma \propto \Pi_n \exp(\Delta S) = \exp(n\,\Delta S)$. Or one can equivalently think of comparing the Euclidean action of $n$ SdS copies to the Euclidean action of $n$ copies of vacuum de Sitter space. The total probability for de Sitter to fragment into $n$ copies would then involve an integral over all masses. Perhaps this probability to spontaneously create multiple copies of arbitrary mass black holes in de Sitter space can somehow be related to the original perturbed Nariai instanton procedure, since the final (fragmented) de Sitter states are the same.

Related to their potential role in de Sitter fragmentation the SdS instantons might also be of relevance in questions related to the de Sitter entropy and a de Sitter version of the information paradox [26, 27]. Concretely, as for the AdS eternal black hole [76], one would expect the exponentially decaying late-time behavior of correlations (for example, between field operators at the poles) in de Sitter space to end at some very large time-scale and instead reach a constant value (on average), in order to be consistent with the finite number of states suggested by the de Sitter entropy. One could speculate that corrections due to (constrained) de Sitter instantons might give rise to this kind of late-time qualitative behavior. For example, for instantons associated to the de Sitter fragmentation, one could imagine having to sum over all spatially disconnected images. As a result of the nonzero overlap between $n$ future (fragmented) de Sitter states and an initial single de Sitter state, these kind of non-perturbative

corrections can perhaps modify the semi-classical exponential decay of correlators at late times, *i.e.,* providing a lower bound naturally suppressed by the Euclidean action of the relevant SdS instanton. We believe it would be of great interest to study this in more detail.

## Acknowledgments

We thank Lars Aalsma, Ben Freivogel, Andrew Svesko and Erik Verlinde for useful discussions. This work is part of the Delta ITP consortium, a program of the Netherlands Organisation for Scientific Research (NWO) that is funded by the Dutch Ministry of Education, Culture and Science (OCW). EM is supported by Ama Mundu Technologies (Adoro te Devote Grant). MV is supported by the Republic and canton of Geneva and the Swiss National Science Foundation, through Project Grants No. 200020-182513 and No. 51NF40-141869 The Mathematics of Physics (SwissMAP).

## A    Near-de Sitter expansion

In this appendix we study the expansion near pure de Sitter of the Euclidean action and the mass parameter for four- and five-dimensional Schwarzschild-de Sitter space. We initiated this analysis already in four dimensions below Eq. (35) in the main text, but here we provide more details. We define the following expansion

$$r_c = L(1 - \epsilon^2), \qquad \text{or} \qquad \epsilon = \pm\sqrt{1 - r_c/L}, \tag{A.1}$$

where $\epsilon$ is a unitless parameter describing the deviation from the empty de Sitter horizon radius. This expansion is useful for visualizing the extrema of the action and for determining the physical (positive) ranges of the horizon radii (associated to the mass range $0 \leq M \leq M_N$).

### A.1    Four dimensions

In four dimensions the blackening factor $f(r)$ (4) has three roots, given by $r_b$, $r_c$ and $r_a = -(r_b + r_c)$. They are related by Eq. (7), which for $n = 2$ and $d = 4$ is given by

$$r_a^2 + r_b^2 + r_c^2 = 2L^2, \qquad \text{or} \qquad r_b^2 + r_c^2 + r_b r_c = L^2. \tag{A.2}$$

This fixes the dependence on $\epsilon$ for the black hole horizon radius

$$r_b = \frac{L}{2}\left(\epsilon^2 + \sqrt{1 + 6\epsilon^2 - 3\epsilon^4} - 1\right), \tag{A.3}$$

and similarly for the other root $r_a$. The $\epsilon$ dependence of the three roots is shown in Figure 8. As $|\epsilon|$ grows, the cosmological horizon shrinks until it matches the growing black hole horizon at the Nariai value $\epsilon = \pm\sqrt{1 - 1/\sqrt{3}}$. Further increasing $|\epsilon|$ will cause the previously called black hole horizon to grow until it becomes the original de Sitter radius $L$, and simultaneously the cosmological horizon radius shrinks to zero at $\epsilon = \pm 1$. By increasing $|\epsilon|$ beyond one, both horizon radii enter a *non-physical regime*, where the black hole horizon radius exceeds the de Sitter radius $L < r_b \leq 2r_N$ and the cosmological horizon radius becomes negative $r_c < 0$. However, for $|\epsilon| > \sqrt{2}$ we see that $r_b$ becomes smaller than $L$ again and the previously negative root $r_a$ becomes positive, until those two horizons coincide at $\epsilon = \pm\sqrt{1 + 2/\sqrt{3}}$.

Further, from Figure 8 we clearly see there are three degenerate cases for the roots: the first case occurs at $r_c = L/\sqrt{3}$ where $r_b = r_c$ and $M = M_N$, the second case is at $r_c = -L/\sqrt{3}$

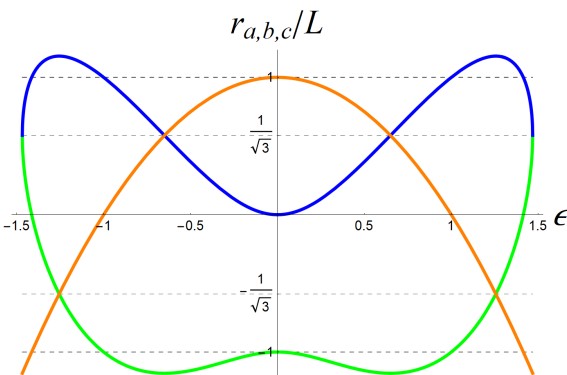

Figure 8: The roots of $f(r)$ in units of $L$ as a function of $\epsilon$ for $d = 4$. The black hole horizon radius $r_b$ is shown in blue, the cosmological horizon radius $r_c$ in orange and the third root $r_a = -(r_b + r_c)$ in green.

where $r_a = r_c$ and $M = -M_N$, and the third case happens at $r_c = -2L/\sqrt{3}$ where $r_a = r_b$ and $M = M_N$. The last case is a new type of Nariai solution.

The fact that for values $|\epsilon| > \sqrt{1 - 1/\sqrt{3}}$ the black hole horizon grows beyond the de Sitter horizon, suggests that they exchange roles. Susskind [26, 27] coined this the "inside-out" process, where the Nariai solution acts as a transition point. However, for $1 < \epsilon < \sqrt{2}$ two roots are negative and the positive one exceeds the original size $L$, thus corresponding to a non-physical regime. Interestingly though, for $|\epsilon| > \sqrt{2}$ two of the roots are positive again and take values between zero and $L$. This is a new physical regime for the SdS black hole, where the root $r_a$ plays the role of the black hole horizon radius and the root $r_b$ becomes the cosmological horizon radius. Beyond $|\epsilon| = \sqrt{1 + 2/\sqrt{3}}$, where the two roots $r_a$ and $r_b$ coincide, these roots become complex.

In order to better understand the (non-)physical regimes we study the mass parameter $M$ as a function of $\epsilon$. Solving $f(r_b) = f(r_c) = 0$ in four dimensions yields

$$M = \frac{r_b}{2G}\left[1 - \left(\frac{r_b}{L}\right)^2\right] = \frac{r_c}{2G}\left[1 - \left(\frac{r_c}{L}\right)^2\right], \tag{A.4}$$

such that in terms of $\epsilon$ the mass takes the form

$$M = \frac{L}{2G}\epsilon^2\left(2 - 3\epsilon^2 + \epsilon^4\right). \tag{A.5}$$

In Figure 9 we plotted $M$ versus $\epsilon$, which shows that the values of $\epsilon$ associated to the non-physical regime correspond to negative values for the mass $M$. On the other hand, the mass is non-negative and takes values in the physical regime $0 \leq M \leq M_N$ for the ranges $0 \leq |\epsilon| \leq 1$ and $\sqrt{2} \leq |\epsilon| \leq \sqrt{1 + 2/\sqrt{3}}$. The latter physical regime has not been considered before in the literature, as far as we are aware, and deserves further investigation, as it is describes a SdS black hole where the horizons correspond to different roots than is usually the case.

In terms of $\epsilon$ the on-shell Euclidean action of SdS, $I_{E,SdS} = -\frac{\pi}{G}\left(r_b^2 + r_c^2\right)$, becomes

$$I_{E,SdS} = -\frac{\pi L^2}{2G}\left(3 - 2\epsilon^2 + \epsilon^4 - (1 - \epsilon^2)\sqrt{1 + 6\epsilon^2 - 3\epsilon^4}\right), \qquad \text{for} \qquad d = 4, \tag{A.6}$$

which is equivalent to Eq. (36). From plotting $I_{E,SdS}$ as a function of $\epsilon$ in Figure 10, we observe that the action has three stationary points: pure de Sitter space is a local minimum, the Nariai black hole a local maximum, and there is a global minimum corresponding to $|\epsilon| = \sqrt{2}$, for which the roots are $r_a = 0, r_b = L, r_c = -L$, resembling pure de Sitter space.

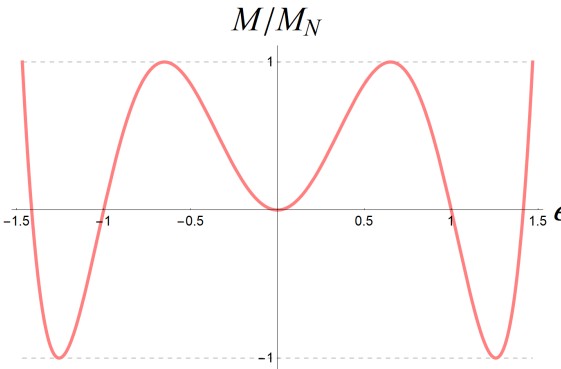

Figure 9: Mass $M$ in units of $M_N$ as a function of $\epsilon$ for $d = 4$. For the values $1 < |\epsilon| < \sqrt{2}$ the mass $M$ becomes negative, and for $|\epsilon| > \sqrt{1 + 2/\sqrt{3}}$ the mass exceeds $M_N$.

### A.2 Five dimensions

In five dimensions the blackening factor $f(r)$ has four roots, given by $r_b$, $r_c$, $r_a = -r_c$ and $r_d = -r_b$. They are related by Eq. (7), which for $n = 2$ and $d = 5$ takes the form

$$r_a^2 + r_b^2 + r_c^2 + r_d^2 = 2L^2, \qquad \text{or} \qquad r_b^2 + r_c^2 = L^2. \tag{A.7}$$

Together with expression (A.1) for $r_c$, this fixes the $\epsilon$ dependence for the other roots of $f(r)$. The black hole horizon radius $r_b$ is given by

$$r_b = L\sqrt{\epsilon^2(2 - \epsilon^2)}, \tag{A.8}$$

and similar expressions exist for $r_a$ and $r_d$. The $\epsilon$ dependence of the four roots is shown in Figure 11. As $|\epsilon|$ grows, the cosmological horizon shrinks until it matches the growing black hole horizon at $\epsilon = \pm\sqrt{1 - 1/\sqrt{2}}$. As $|\epsilon|$ increases, the "black hole" horizon reaches $r_b = L$, where $r_c = 0$ and $\epsilon = \pm 1$. Beyond that $r_b$ begins to shrink and $r_a$ becomes positive, until they meet at $\epsilon = \pm\sqrt{1 + 1/\sqrt{2}}$. Afterwards $r_b$ shrinks back to zero at $\epsilon = \pm\sqrt{2}$ where $r_a = L$. For $|\epsilon| > \sqrt{2}$, $r_a$ and $r_b$ become complex, whereas $r_c = -r_a < -L$. Further, from Figure 11 we see there are five degenerate cases for the roots: i) $r_b = r_d = 0$ at $\epsilon = 0$, ii) $r_a = r_d = -r_N$ and $r_b = r_c = r_N$ at $|\epsilon| = \sqrt{1 - 1/\sqrt{2}}$, iii) $r_a = r_c = 0$ at $|\epsilon| = 1$, iv) $r_a = r_b = r_N$ and $r_c = r_d = -r_N$ at $|\epsilon| = \sqrt{1 + 1/\sqrt{2}}$, and v) $r_b = r_d = 0$ at $|\epsilon| = \sqrt{2}$.

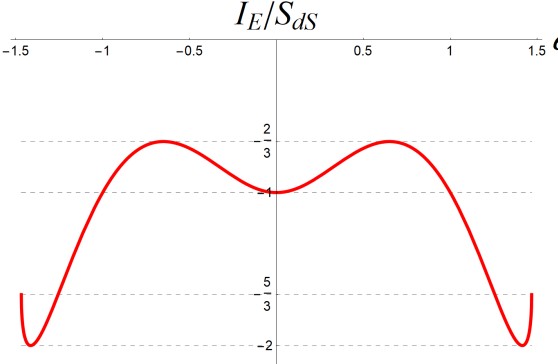

Figure 10: Euclidean action $I_E$ of SdS in units of $S_{dS}$ as a function of $\epsilon$ for $d = 4$. Empty de Sitter is a minimum and Nariai a maximum of the action, and there is an additional global minimum at $|\epsilon| = \sqrt{2}$ for which $M = 0$.

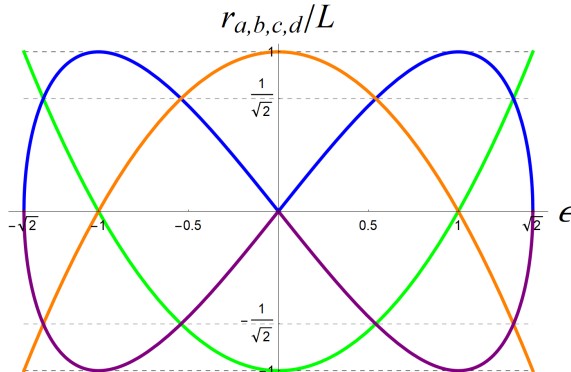

Figure 11: The roots of $f(r)$ in units of $L$ as a function of $\epsilon$ for $d = 5$. The black hole horizon radius $r_b$ is shown in blue, the cosmological horizon radius $r_c$ in orange, the root $r_a = -r_c$ in green and the fourth root $r_d = -r_b$ in purple.

In five dimensions we notice that the roots never exceed $L$ and that the number of positive and negative roots stays constant throughout, indicating that the horizons never enter a non-physical regime. Studying the corresponding mass as a function of $\epsilon$ confirms this, as in five dimension the mass is

$$M = \frac{3\pi r_b^2}{8G}\left[1 - \left(\frac{r_b}{L}\right)^2\right] = \frac{3\pi r_c^2}{8G}\left[1 - \left(\frac{r_c}{L}\right)^2\right], \tag{A.9}$$

and in terms of $\epsilon$ it is given by

$$M = \frac{3\pi L^2}{8G}\epsilon^2\left(2 - \epsilon^2\right)\left(1 - \epsilon^2\right)^2. \tag{A.10}$$

In Figure 12 we plotted the mass as a function of $\epsilon$, and used that the Nariai mass is given by $M_N = 3\pi L^2/32G$ in five dimensions. As opposed to the four-dimensional case, defining a unitless parameter $\epsilon$ as the deviation from the de Sitter cosmological horizon radius, will not cause the mass to enter a non-physical regime $M < 0$. This pattern continues in higher dimensions, in the sense that in even dimensions one enters a non-physical regime with $M < 0$, whereas in odd dimensions one always remains in the physical regime $0 \leq M \leq M_N$.

The on-shell Euclidean action, $I_{E,SdS} = -(A_b + A_c)/4G$, can be computed to be

$$I_{E,SdS} = -\frac{\pi^2 L^3}{2G}\left((1 - \epsilon^2)^3 + \left(\epsilon^2(2 - \epsilon^2)\right)^{3/2}\right), \qquad \text{for} \qquad d = 5. \tag{A.11}$$

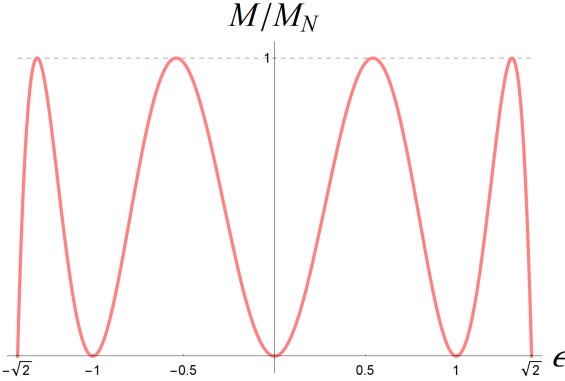

Figure 12: Mass in units of $M_N$ as a function of $\epsilon$ for $d = 5$.

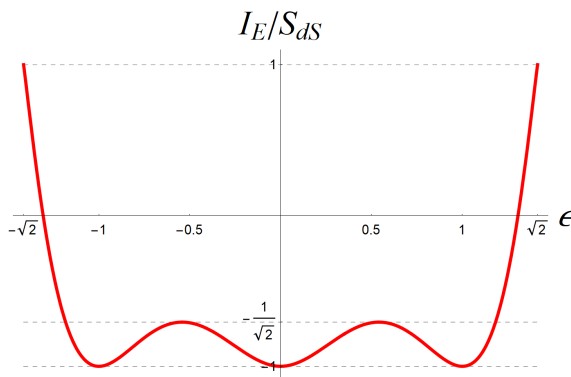

Figure 13: Euclidean action in units of $S_{dS}$ as a function of $\epsilon$ for $d = 5$.

From Figure 13 we see that the action has again three stationary points: de Sitter space at $\epsilon = 0$, the Nariai black hole at $|\epsilon| = \sqrt{1 - 1/\sqrt{2}}$, and the de Sitter-like solution at $|\epsilon| = 1$.

## B    Near-Nariai expansion

In this appendix we expand the Euclidean action around the Nariai solution, restricting to four dimensions. In the main text, below Eq. (37), we used the expansion parameter $\tilde{\epsilon}$ defined as

$$r_c = r_N(1 + \tilde{\epsilon}). \tag{B.1}$$

The black hole horizon radius is fixed by the relation (A.2 )

$$r_b = \frac{r_N}{2} \left( \sqrt{3(1 - \tilde{\epsilon})(3 + \tilde{\epsilon})} - (1 + \tilde{\epsilon}) \right), \tag{B.2}$$

where we inserted $L^2 = 3r_N^2$. In terms of $\tilde{\epsilon}$ the on-shell action of SdS, $I_{E,SdS} = -\frac{\pi}{G}\left(r_b^2 + r_c^2\right)$, becomes

$$I_{E,SdS} = -\frac{\pi r_N^2}{2G} \left( 7 + \tilde{\epsilon}(2 + \tilde{\epsilon}) - (1 + \tilde{\epsilon})\sqrt{3(1 - \tilde{\epsilon})(3 + \tilde{\epsilon})} \right), \tag{B.3}$$

which agrees with Eq. (39). To second order in $\tilde{\epsilon}$ around the Nariai geometry the action is

$$I_{E,nN} = -\frac{2\pi r_N^2}{G}\left(1 + \frac{2}{3}\tilde{\epsilon}^2\right) + \mathcal{O}(\tilde{\epsilon}^3) = -\frac{2\pi}{\Lambda G} - \frac{4\pi}{3\Lambda G}\tilde{\epsilon}^2 + \mathcal{O}(\tilde{\epsilon}^3). \tag{B.4}$$

This shows there exists a negative mode for the near-Nariai solution. However, the precise coefficient of the negative mode in the action is not the same as in previous literature. In Eq. (3.11) of [9] Ginsparg and Perry obtained $-\frac{20\pi}{9\Lambda G}\epsilon_0^2$ for the negative mode, whereas in Eq. (A.18) of [10] Bousso and Hawking found $-\frac{17\pi}{9\Lambda G}\epsilon_0^2$. We should add though that our expansion parameter $\tilde{\epsilon}$ is different from the parameter $\epsilon_0$ used by these authors, and hence one might expect a different result. In the following we will redo the computation of the on-shell action using the expansion parameter $\epsilon_0$, defined below, and show that the result remains the same as (B.4 ). Thus, our result for the negative mode is in disagreement with [9,10].

In the Appendix of [10] Bousso and Hawking used the definition for the expansion parameter $\epsilon_0$ around the Nariai mass as introduced by Ginsparg and Perry [9][6]

$$\left(\frac{M}{M_N}\right)^2 = 9G^2M^2\Lambda = 1 - 3\epsilon_0^2. \tag{B.5}$$

---

[6]In arbitrary dimensions this can be generalized as $(M/M_N)^2 = 1 - (d-1)(d-3)\epsilon_0^2$.

However, Bousso and Hawking corrected the near-Nariai expansion in [9] by making sure that $f(r) = 0$ on the horizons, and by properly normalizing the Killing vector. Condition (A.2) ensures that $f(r) = 0$ on the horizon, and can be used to find the expansion of the horizon radii

$$r_{b,c} = \frac{1}{\sqrt{\Lambda}}\left(1 \mp \epsilon_0 - \frac{1}{6}\epsilon_0^2 \mp \frac{4}{9}\epsilon_0^3\right) + \mathcal{O}(\epsilon_0^4). \tag{B.6}$$

Next, they define new time and radial coordinates $\psi$ and $\chi$ by[7]

$$\tau = \frac{1}{\epsilon_0\sqrt{\Lambda}}\left(1 - \frac{1}{2}\epsilon_0^2\right)\psi, \qquad r = \frac{1}{\sqrt{\Lambda}}\left[1 + \epsilon_0\cos\chi - \frac{1}{6}\epsilon_0^2 + \frac{4}{9}\epsilon_0^3\cos\chi\right]. \tag{B.7}$$

The black hole horizon is located at $\chi = \pi$ and the cosmological horizon is at $\chi = 0$. The radial coordinate is determined by (B.6), and the time coordinate follows from the definition of the Killing vector $\xi = \frac{1}{\sqrt{f(r_0)}}\partial_\tau = \sqrt{\Lambda}\partial_\psi$, which is normalized such that it has unit length on the geodesic where an observer needs no acceleration to stay in place. The near-Nariai metric in Euclidean signature is up to second order in $\epsilon_0$

$$ds^2 = \frac{1}{\Lambda}\left(1 - \frac{2}{3}\epsilon_0\cos\chi + \frac{2}{3}\epsilon_0^2\cos^2\chi - \frac{1}{9}\epsilon_0^2\right)\sin^2\chi\, d\psi^2 + \frac{1}{\Lambda}\left(1 + \frac{2}{3}\epsilon_0\cos\chi - \frac{2}{9}\epsilon_0^2\cos^2\chi\right)d\chi^2$$

$$+ \frac{1}{\Lambda}\left(1 + 2\epsilon_0\cos\chi + \epsilon_0^2\cos^2\chi - \frac{1}{3}\epsilon_0^2\right)d\Omega_2^2. \tag{B.8}$$

The surface gravities associated to $\xi$ for the two horizons were computed in Eq. (18)

$$\kappa_{b,c} = \sqrt{\Lambda}\left(1 \pm \frac{2}{3}\epsilon_0 + \frac{11}{18}\epsilon_0^2\right). \tag{B.9}$$

To remove the conical singularity at one of the horizons one needs the periodicity

$$\psi_{b,c}^{id} = \frac{2\pi\sqrt{\Lambda}}{\kappa_{b,c}} = 2\pi\left(1 \mp \frac{2}{3}\epsilon_0 - \frac{1}{6}\epsilon_0^2\right). \tag{B.10}$$

The Euclidean action of SdS can now be computed as follows

$$I_E = I_{E,bulk} + I_{E,con,b} + I_{E,con,c} = -\frac{\mathcal{V}\Lambda}{8\pi G} - \frac{A_b\epsilon_b}{8\pi G} - \frac{A_c\epsilon_c}{8\pi G}. \tag{B.11}$$

Here $\mathcal{V}$ is the four-volume of the Euclidean singular geometry, and $\epsilon_{b,c}$ are the conical deficit angles at the horizons. For an *arbitrary* periodicity $\psi^{id}$ of the Euclidean time $\psi$, the conical deficit angles are $\epsilon_{b,c} = 2\pi - \psi^{id}\kappa_{b,c}/\sqrt{\Lambda}$, cf. the discussion below Eq. (29). Using the near-Nariai metric (B.8) and the surface gravities (B.9) the three terms can be shown to be

$$I_{E,bulk} = -\frac{\psi^{id}}{\Lambda G} + \frac{\psi^{id}}{18\Lambda G}\epsilon_0^2 + \mathcal{O}(\epsilon_0^3), \tag{B.12}$$

$$I_{E,con,b} = \frac{\psi^{id} - 2\pi}{2\Lambda G} - \frac{(2\psi^{id} - 6\pi)}{3\Lambda G}\epsilon_0 - \frac{(\psi^{id} + 24\pi)}{36\Lambda G}\epsilon_0^2 + \mathcal{O}(\epsilon_0^3), \tag{B.13}$$

$$I_{E,con,c} = \frac{\psi^{id} - 2\pi}{2\Lambda G} + \frac{(2\psi^{id} - 6\pi)}{3\Lambda G}\epsilon_0 - \frac{(\psi^{id} + 24\pi)}{36\Lambda G}\epsilon_0^2 + \mathcal{O}(\epsilon_0^3). \tag{B.14}$$

Finally, by adding up the three terms we find that $\psi^{id}$ drops out in the total action

$$I_{E,nN} = -\frac{2\pi}{\Lambda G} - \frac{4\pi}{3\Lambda G}\epsilon_0^2 + \mathcal{O}(\epsilon_0^3). \tag{B.15}$$

---

[7]In general dimensions the time coordinate is defined by $\tau = \frac{r_N}{\epsilon}\left(\frac{1}{d-3} - \frac{d-2}{4}\epsilon_0^2\right)\psi$ and the radial coordinate is defined via $r = r_N\left(1 + \epsilon_0\cos\chi - \frac{1}{6}(2d-7)\epsilon_0^2 + \frac{1}{18}(5d^2 - 26d + 32)\epsilon_0^3\cos\chi\right)$.

Although we were able to reproduce the near-Nariai metric and surface gravities in [10], we obtained a different coefficient for the $\epsilon_0^2$ term in the action. In fact, we find the same coefficient as in Eq. (B.4), thus confirming our result for the negative mode. To be clear, we agree with all the equations in the Appendix of [10], except for the final result (A.18) for the action. Curiously, they find a different result depending on the value for the periodicity $\psi^{id}$. In particular, for $\psi^{id} = \psi^{id}_{b,c}$ they obtained $-\frac{17\pi}{9\Lambda G}\epsilon_0^2$ and for $\psi^{id} = 2\pi$ they found $-\frac{20\pi}{9\Lambda G}\epsilon_0^2$ for the negative mode. In the main text, however, we showed that the total Euclidean action of SdS is independent of the time periodicity, and Eq. (B.15) is a nontrivial check of that result.

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
