# Peer review of "On the Euclidean Action of de Sitter Black Holes and Constrained Instantons"

_SciPost Physics, doi:SciPost Phys. 14, 022 (2023)_

## Round 2 · Referee Report · Anonymous (Referee 1) · 2022-5-10

Strengths

Schwarzschild de Sitter black holes are a well-studied system, but the authors have been able to uncover two new and surprising features of the Euclidean solutions: 1. The vanishing of the term linear in beta in the Euclidean action. 2. The approximate linear dependence of the entropy difference on mass.

Weaknesses

I would appreciate a fuller discussion of the application of the constrained instanton method to these solutions. The authors point to the earlier paper [24] for more detail, but as this is an important part of their story, I think more explanation here would be welcome.

Report

I am happy to recommend publication- the authors have found some interesting results, and this work has the potential to reinvigorate the exploration of black holes in de sitter as a probe of its microscopic description.

Requested changes

  1. On the constrained instantons, I would appreciate in particular an explanation of the claim on page 16 that the solutions can be understood as smooth in this context.

  • validity: high
  • significance: high
  • originality: high
  • clarity: high
  • formatting: perfect
  • grammar: excellent

Author:  Jan Pieter van der Schaar  on 2022-07-15  [id 2663]

(in reply to Report 1 on 2022-05-10)
Category:
remark
answer to question

First, let me apologise for the long delay in our reply to the referee report, which to some extent can be explained by ongoing follow-up work that actually pertains to the 'weakness' pointed out by the referee. We thank the referee for the otherwise mostly positive comments.

With regards to the identified weakness we agree that a fuller discussion of the application of constrained instantons would be beneficial and intend to update the manuscript accordingly, as soon as possible, in conjunction with our follow-up work. Specifically, with respect to the requested change, we will clarify or rephrase our supposed claim regarding the smooth nature of the solutions.

We expect to upload an updated version of the manuscript, addressing these points, after the Summer break.

---

## Round 2 · Referee Report · Anonymous (Referee 2) · 2022-7-27

Strengths

  1. Presents a new approximate linear relation between the difference of the entropies of Schwarschild de Sitter and de Sitter, and the mass of the black hole.
  2. Derives the probability of BH pair creation in a novel manner, using the formalism of constrained instantons, and finds qualitatively similar results to previous literature.

Weaknesses

  1. In the introduction, the derivation of equation (1.1) is presented as the main result of the paper, although this result was already reported in the reference [27]. I think this is a bit misleading.
  2. The implementation of the constraints on the metric, through the constraint functional C[g], is not computed though it seems like a crucial part of the constrained instantons formalism.

Report

This submission presents some original results on the Schwarschild-de Sitter geometry, and make use of the constrained instantons formalism, showing it leads to expected results at least qualitatively and hence bringing support to this method for computing gravitational path integral contributions. In my opinion, it suits the journal criteria and I recommend publication after the few requested changes points have been addressed.

Requested changes

  1. First a typo: I think following its caption, the vertical axis on figure 6 should read $S_{SdS}/S_{dS}$ instead of the current legend.

  2. A small comment: in the abstract and in section 2.2, the authors state that despite the conical singularities present in the SdS spacetime, the on-shell action turns out to be finite. I'd like to point out the following papers: Barrow and Tipler, Nature 331 (1988) 31; Barrow 2019, arxiv: 1912.12926. There it was found that spacetimes with such a singularity have rather generically a finite on-shell action; this was even proposed as a trade-off situation, where one either gets an infinity in the on-shell action, or a singularity in the spacetime. Additionally, I find that the denomination as “mild singularities” on page 9 is not precise, what do the authors mean technically? Is it that the singularities are simple poles and not essential singularities?

  3. Just for the sake of completeness, I wonder if the authors also tried to fit other kinds of functions than the linear function for the difference of entropies between vacuum dS and the SdS solution, in order to compare how good the fits are, or if they have an argument for why this is not needed (except for the fact that the relative difference is of the order of the percent).

  4. In Appendix B the authors show that their calculation of the negative mode for the SdS solution in the regime called "near-Nariai", close to the solution with maximal mass, gives as in previous literature a negative mode in direction of decreasing black hole mass, but not the same quantitative coefficient for this negative mode. Do the authors have an idea what this discrepancy is due to?

  5. Even if it is stated at the end of the introduction that [27] contains "similar results for the on-shell action of Schwarschild-de Sitter", I believe it should be cited before, when those results are stated (in (1.1) for example)

  • validity: high
  • significance: good
  • originality: good
  • clarity: high
  • formatting: excellent
  • grammar: good

Author:  Jan Pieter van der Schaar  on 2022-09-20  [id 2835]

(in reply to Report 2 on 2022-07-27)
Category:
answer to question

We thank the referee for the critical, positive, comments. We have prepared a revised version addressing the requested changes as best as we could, specifically:

  1. We fixed the typo in the caption of figure 6.

  2. We agree that the term 'mild singularities' on page 9 is vague and rephrased what we actually mean. Regarding the papers that were pointed out by the referee: although they are certainly interesting, we do not believe they are relevant to the specific Euclidean de Sitter problem we are studying.

  3. We added a footnote in the relevant section, explaining that the linear fit can be understood as a large d expansion and that we neglected higher order terms, de facto also pointing out that we did not try another fit.

  4. We checked once more what we wrote in appendix B, and although we were not explicit, out of politeness, we think it is pretty clear from the context that we believe Bousso and Hawking made a mistake. So we are inclined not to make any changes here.

  5. We agree, and we have moved the citation to the right place.

We hope to have responded appropriately to the referee's requests, and expect to upload a revised version for publication soon.

Author:  Jan Pieter van der Schaar  on 2022-09-13  [id 2810]

(in reply to Report 2 on 2022-07-27)
Category:
answer to question
correction

We thank the referee for the critical, positive, comments. We have prepared a revised version addressing the requested changes as best as we could, specifically:

  1. We fixed the typo in the caption of figure 6.
  2. We agree that the term 'mild singularities' on page 9 is vague and rephrased what we actually mean. Regarding the papers that were pointed out by the referee: although they are certainly interesting, we do not believe they are relevant to the specific Euclidean de Sitter problem we are studying.
  3. We added a footnote in the relevant section, explaining that the linear fit can be understood as a large d expansion and that we neglected higher order terms, de facto also pointing out that we did not try another fit.
  4. We checked once more what we wrote in appendix B, and although we were not explicit, out of politeness, we think it is pretty clear from the context that we believe Bousso and Hawking made a mistake. So we are inclined not to make any changes here.
  5. We agree, and we have moved the citation to the right place.

We hope to have responded appropriately to the referee's requests, and expect to upload a revised version for publication soon.

---

## Round 3 · Referee Report · Anonymous (Referee 1) · 2022-10-7

Report

The authors have satisfactorily responded to the comments in previous reports; I am happy to recommend publication.

---

## Round 3 · Referee Report · Anonymous (Referee 2) · 2022-10-13

Report

The weaknesses raised by last report were satisfactorily replied to. I recommend publication.

---

## Round 3 · Author Response

In this revised version we address all the (small) issues raised by the referees, which we now expect to be ready for publication.

---

## Round 3 · List of Changes

• We have moved some citations in the introduction to the more appropriate place.
  • We have added several small clarifications and slightly improved the writing in several places
  • We fixed a typo in the caption of figure 6.
  • We removed the term 'mild singularities' on page 9 and rephrased what we actually mean.
  • We extended the discussion of the application of constrained instantons. In particular, we have also clarified our remark regarding the smooth nature of the solutions.
  • We added a footnote explaining that the linear fit can be understood as a large d expansion and that we neglected higher order terms.
  • We have added some references

---

## Editorial Decision

published